# Serum Proteomics in Patients with Head and Neck Cancer: Peripheral Blood Immune Response to Treatment

**DOI:** 10.3390/ijms23116304

**Published:** 2022-06-04

**Authors:** Thorsteinn Astradsson, Felix Sellberg, Ylva Tiblom Ehrsson, Karl Sandström, Göran Laurell

**Affiliations:** 1Department of Surgical Sciences, Uppsala University, 751 85 Uppsala, Sweden; ylva.tiblom.ehrsson@surgsci.uu.se (Y.T.E.); karl.sandstrom@surgsci.uu.se (K.S.); goran.laurell@surgsci.uu.se (G.L.); 2Department of Immunology, Genetics and Pathology, Uppsala University, 751 85 Uppsala, Sweden; felix.sellberg@igp.uu.se

**Keywords:** cytokines, protein expression, chemoradiotherapy, oropharyngeal cancer, cisplatin

## Abstract

In this real-world study, the aims were to prospectively evaluate the expression of inflammatory proteins in serum collected from head and neck cancer patients before and after treatment, and to assess whether there were differences in expression associated with treatment modalities. The mixed study cohort consisted of 180 patients with head and neck cancer. The most common tumor sites were the oropharynx (*n* = 81), the oral cavity (*n* = 53), and the larynx (*n* = 22). Blood tests for proteomics analysis were carried out before treatment, 7 weeks after the start of treatment, and 3 and 12 months after the termination of treatment. Sera were analyzed for 83 proteins using an immuno-oncology biomarker panel (Olink, Uppsala, Sweden). Patients were divided into four treatment groups: surgery alone (Surg group, *n* = 24), radiotherapy with or without surgery (RT group, *n* = 94), radiotherapy with concomitant cisplatin (CRT group, *n* = 47), and radiotherapy with concomitant targeted therapy (RT Cetux group, *n* = 15). For the overall cohort, the expression levels of 15 of the 83 proteins changed significantly between the pretreatment sample and the sample taken 7 weeks after the start of treatment. At 7 weeks after the start of treatment, 13 proteins showed lower expression in the CRT group compared to the RT group. The majority of the inflammatory proteins had returned to their pretreatment levels after 12 months. It was clearly demonstrated that cisplatin-based chemoradiation has immunological effects in patients with head and neck cancer. This analysis draws attention to several inflammatory proteins that are of interest for further studies.

## 1. Introduction

Head and neck cancer (HNC) is a collection of tumors that originate in different locations in the oral cavity, pharynx, sinonasal tract, larynx, and salivary glands. HNC is the sixth most common cancer type worldwide, and its incidence has increased, mainly due to an increase in HPV-positive oropharyngeal cancer [1]. Squamous-cell carcinoma is the dominant histology, but a wide variety of histological types can be identified. 

In the past few decades, the focus has been on the immunological nature of oncogenesis and treatment. Although a large number of experimental studies have been performed, the clinical data are relatively unclear regarding the immunology of HNC compared to that of many other solid malignant tumors. Little is known about the systemic immune response to treatment, and there is a growing need for more translational research [2].

It has become clear that cancer cells exploit the immune system when establishing a tumor microenvironment (TME) that facilitates their progression [3]. This immune evasion involves, for instance, defective antigen presentation, the upregulation of T-regulatory cells, and the secretion of immunosuppressive mediators. 

Understanding the interaction between the tumor and the immune system is crucial to establish more effective treatment options. To date, significant progress has been made in immunotherapy approaches for malignant melanoma and lung cancer. Furthermore, the identification of cytokines and other molecules is important, as it can contribute to the diagnosis of cancer, and may have prognostic value. Systemic inflammation markers have been shown to be useful in predicting the prognosis of various types of cancer, including HNC [4,5,6]. Refinement of immunological pathway assessment methods could be of immense value as a prognostic tool. 

Radiotherapy (RT) is a mainstay in HNC treatment, in combination with surgery and/or chemotherapy. Ionizing radiation exerts its effect on cancer cells through DNA damage, which induces cell death and tumor regression. It is also evident that radiation induces an immune reaction in the TME, which is of great importance for the pro-inflammatory or anti-inflammatory response [7]. This activation of the immune system can trigger the eradication of tumor cells through various mechanisms [8,9]. RT also has the potential to act synergistically with immunotherapy agents through checkpoint inhibitors. 

The mechanisms by which radiation therapy activates the local immune response are complex, and include the presentation of neoantigens and the promotion of T-cell infiltration into the TME. Put simply, RT thus has the potential to turn so-called immunologically ‘cold’ tumors into ‘hot’ tumors, and subsequently expose cancer cells to the immune system [10,11]. 

Cisplatin is widely used in the treatment of HNC, and has been shown to increase tumor immunogenicity, with the potential to enhance the effects of immunotherapy agents [12]. In vitro studies have shown that low-dose cisplatin activates and promotes the recruitment of cytotoxic T cells to tumor sites [13,14]. Cisplatin upregulates the expression of MHC class I in cancer cells, enabling their identification by the immune system while also counteracting the immunosuppressive environment that cancer cells promote [15].

In a previously published exploratory study, we showed that patients undergoing HNC treatment—and especially patients who developed recurrence—exhibited a spike in pro-inflammatory cytokines in their sera 7 weeks after the initiation of treatment [16]. In the present study, our objective was to further explore the systemic immune response in a larger cohort of patients with HNC through the longitudinal analysis of cytokines and different proteins known to be involved in tumor immunity, chemotaxis, vascularization, and tissue remodeling. More specifically, we wanted to investigate, in a real-world study, whether the expression of serum inflammatory proteins varied over time, and whether there were differences in expression associated with treatment modalities.

## 2. Results

### 2.1. Protein Expression over Time

A total of 669 blood samples was analyzed for protein expression (NPX) values over time. Blood samples from all four time points were available from 127 patients, three blood samples from 36 patients, two blood samples from 12 patients, and one blood sample from 5 patients. Most changes in protein expression were observed with respect to treatment. As shown in Table 1, for the whole cohort, the expression levels of 15 of the 83 proteins changed significantly between the pretreatment sample and the sample taken 7 weeks after the start of treatment. For patients who received RT, this time point corresponded to the end of treatment. Several proteins also changed significantly at 3 months after the end of treatment, as shown in Table 1. The CCL20 and ARG1 concentrations decreased and the CCL17 concentration increased at the three time points compared to baseline. 

### 2.2. Treatment Modalities and Effects on Protein Expression

Out of the 94 patients in the RT group, 50 were treated with single-modality radiotherapy, 7 received preoperative radiotherapy, and 37 underwent postoperative radiotherapy. In the CRT group, 31 of the 47 patients were treated with chemoradiation with concomitant cisplatin, 6 patients received chemoradiation after primary surgery, and 8 patients were treated with neck dissection after an assessment by PET/CT approximately 3 months after the termination of treatment. Furthermore, 13 of the 15 patients in the RT Cetux group received radiotherapy with concomitant cetuximab, and 2 patients received surgical intervention prior to the oncological treatment.

From the results shown in Figure 1, it is evident that the levels of inflammatory proteins varied between treatment groups, and also between times of follow-up. 

We investigated whether protein expression differed between the four treatment groups (Surg, RT, CRT, and RT Cetux). Analysis of the 83 proteins showed that 17 proteins exhibited significant changes between the treatment groups at 7 weeks. These were further analyzed by multivariate analysis including the variables tumor stage, mucositis, smoking pack-years, and change in body weight. For 14 out of these 17 proteins, the treatment modality retained the strongest correlation with protein expression. The patterns of these proteins are presented in Figure 2. 

Other correlations identified in the multivariate analysis can be found in Table 2. Correlations with proteins originally identified in the two-way ANOVA, in which the correlation was stronger with other variables, are indicated in bold *(n* = 3). The sample size of patients with p16-negative oropharynx cancer was too small; thus, p16 was not included in the analysis. Changes in IL-6 and IL-10 concentrations were significantly associated with mucositis, body weight change, and smoking pack-years, but not with the modality of treatment or tumor stage. Conversely, the expression of the FAS ligand and IL-12 demonstrated significant associations with both types of RT with concomitant pharmacological treatment (cisplatin or cetuximab). The decrease in the level of CD5 concentration depended on RT and RT with concomitant pharmacological treatment.

The differences between the treatment groups waned at 12 months after the end of treatment. 

## 3. Discussion

In this real-world study prospectively evaluating a mixed HNC population, comparative inflammatory protein data were analyzed for patients assigned to four treatment groups. With 180 patients, this study is, to the best of our knowledge, the largest prospective longitudinal investigation of serum inflammatory proteins in patients with HNC. An overwhelming majority of the participants were diagnosed with squamous-cell carcinoma (93%), most commonly in the oral cavity and oropharynx (74% of all patients). 

The main findings include the significant association between inflammatory proteins and treatment modalities. Compared to RT with or without surgical invention, RT with concomitant cisplatin induced a significant decrease in the levels of 13 of the 83 serum proteins analyzed at the end of treatment.

The chain of immune reactions and the trafficking of inflammatory markers among various cells in the tumor compartments can be defined in biopsies and surgical specimens, and used for analyses of the tumor tissue and tumor microenvironment (TME). The blood sampling used in the present study was a minimally invasive method to detect, quantify, and monitor changes in the blood compartment in patients with malignant tumors, offering the possibility of repeated measurements after the termination of treatment. Although a growing number of preclinical proteomics studies have characterized many pathways involved in the interaction and communication between tumor tissue and the TME, little knowledge is available about the interplay between these compartments and peripheral blood. 

### 3.1. Protein Expression over Time

The present study identified a fraction of proteins in the blood compartment that were of specific interest. For the whole cohort, a clear time pattern was evident among several serum inflammatory proteins, with the greatest changes seen 7 weeks after the initiation of treatment. The expression levels of 15 proteins changed significantly between pretreatment and the seven-week time point—the period corresponding to the termination of the first-line treatment. Of these, 5 proteins demonstrated increased expression levels and 10 proteins showed decreased expression levels. At 3 months after the end of treatment, 11 proteins returned to pretreatment levels. Three proteins remained stable during repeated sampling: the expression levels of CCL20 and ARG1 were lower, and the expression level of CCL17 was higher, compared to the pretreatment level, at all three time points after treatment. It can be hypothesized that several mechanisms were responsible for the downregulation and upregulation of these proteins. CCL20 is also called macrophage inflammatory protein-3 [17]. It has recently been reported that CCL20 in tumor-bearing mice can turn a cold tumor into a hot tumor, and promote tumor cell elimination [18]. It can be assumed that infiltration of CCL20 into the TME from the blood compartment could play a vital role in tumor eradication, thereby affecting the prognosis. CCL17 is known to be released by tumor cells and tumor-associated macrophages, and promotes tumor development [19]. However, levels of CCL17 may represent a marker of interest to study in HNC, as increased serum levels have been shown to be associated with longer survival in melanoma patients with metastasis [20].

### 3.2. Treatment Modality and Effects on Protein Expression

We identified differences in the serum levels of inflammatory proteins between the treatment groups, with a clear pattern of changes in the CRT and RT Cetux groups at the seven-week and three-month time points. Surgery alone did not affect the expression levels of serum inflammatory proteins. A strong influence of treatment modality on serum inflammatory protein expression levels was identified at 7 weeks in the CRT group, where expression levels were significantly lower in 13 proteins, which could be correlated with cisplatin treatment. This finding strengthens the evidence that cisplatin, in addition to its cytotoxic effects, also induces an immunomodulatory response in connection with treatment in patients with oropharyngeal cancer. Measurements of serum expression levels in cancer patients undergoing cisplatin treatment have earlier been explored for four of these inflammatory proteins, namely, FASLG, CXCL5, CD5, and IL-12 [21,22,23,24]. Thus, we identified decreased expression levels of nine additional inflammatory proteins in sera. 

CD5 was lower in all three groups receiving radiotherapy compared to the Surg group at all three follow-up time points. Downregulated expression of CD5 in tumor-infiltrating lymphocytes has been shown to provide antitumor effects in patients with lung cancer, and the decrease in serum concentration seen in the present study could reflect this as well [25]. However, the role of soluble CD5 in this HNC cohort remains elusive. 

The protein profile was clearly shown to be different in the RT Cetux group compared to the CRT group. While most changes in the CRT group were noted at 7 weeks after the start of treatment, some of the greatest changes in the RT Cetux group were evident at 3 months after the termination of treatment.

Cisplatin is widely studied, and is known to be extremely reactive and cause cancer cell death through two main mechanisms: interaction with DNA, and mitochondrial damage causing oxidative stress [26]. Mitochondrial damage has been associated with several side effects induced by cisplatin, such as peripheral neuropathy, ototoxicity, and nephrotoxicity, to which the release of FASL is likely to contribute [27,28,29]. The human clinical data obtained in the present study show that after almost 50 years of intensive research on cisplatin’s effects, we are still learning new ways in which this substance exerts influence on the host and cancer.

Previous studies have shown that different mechanisms are responsible for the immunomodulatory effects of cisplatin, of which downregulation of immunosuppressive TME is crucial [15,30]. Animal research has provided a more extensive explanation of how cisplatin affects tumor immunity. Nejad et al. reported in an in vitro and in vivo murine model that cisplatin administered as monotherapy acts through the induction of tumor-specific CD8+ cells [31]. Furthermore, cell death induced the release of 19 proteins in the TME. Several studies have also used preclinical HNC models to identify inflammatory biomarkers related to cisplatin treatment. For example, the expression of MMP7 and MMP 13 in HNC cell lines has been shown to be associated with resistance to cisplatin [32]. 

Preclinical research—mainly in murine models—has also been used to explore the immunogenic properties of cisplatin in combination with radiation therapy [30,33,34]. The combination of cisplatin and irradiation has been shown to produce increased apoptosis and reactive oxygen species (ROS) activity in tumor tissue. Taken together, the results from the present cohort and, in particular, the CRT group, indicate that several mechanisms may be involved in the combined cytotoxic effect, as shown by the immuno-oncological protein profiles in serum. 

Among the proteins downregulated in the CRT group 7 weeks after the start of treatment were FASLG, IL-12, TNFSF14, and CD244. Three of these proteins—IL 12, TNFSF14, and CD244—have previously been considered as useful for clinical application in cancer therapy as promoters of tumor cell apoptosis, or as therapeutic targets [35,36,37]. 


*FAS/FASL*


FAS is a transmembrane protein that belongs to the TNF superfamily, which contains key mediators of the immune system. When activated by its ligand (FASL), FAS induces apoptosis, and can therefore be considered a marker of cell apoptosis, including that of cancer cells [21]. It is reported that a low FASL/FAS ratio in tumors is a negative prognostic marker, and the administration of tumor-targeted antibodies fused to FASL could be a future therapeutic alternative in such patients [38,39,40]. Hoffman et al. reported that serum levels of soluble FASL (sFASL) were lower in HNC patients with active disease than in tumor-free patients, due to spontaneous apoptosis of T cells [41]. However, another clinical study did not find any association between sFASL and disease activity in patients with oral cancer [42]. Little is known about the role of cisplatin in sFASL and FAS-mediated apoptosis. Here, we observed significantly decreased levels of FASL in the CRT group at 7 weeks after the start of treatment compared to the other treatment groups. This finding gives some insight into how cisplatin modulates the FASL/FAS effector mechanism in HNC chemoradiation therapy. According to previous research, a low FASL concentration could be a sign of resistance of the cancer cells to cisplatin [43].


*IL-12*


IL-12 is an immune-activating cytokine, and has been shown to regulate NK cells and CD8 T cells [44]. Several inflammatory cytokines, such as IL 10, are known to inhibit IL-12 production and, therefore, affect the antitumor cell-mediated response (Th1 response) [45]. Infusion of IL-12 in a murine model has been shown to induce increased antitumor immunity, with increased infiltration of NK cells and CD8 cells in the tumor compartment [46]. Other animal models have also demonstrated a potent antitumor effect of IL-12, but this has not been observed in humans so far [46,47]. Serum levels of IL-12 in cancer patients have been studied for several decades. Jebreel et al. reported that a cohort of 57 HNC patients had lower preoperative levels of IL-12 compared to healthy controls, and concluded that this reflected an immunological environment that favored tumor evolution [48]. Whether the lower levels of IL-12 in the CRT group represent an effect of cisplatin, or more advanced disease in the cisplatin group, remains elusive; however, the latter is somewhat unlikely, as IL-12 levels did not correlate with tumor stage.


*TNFSF14*


TNFSF14 (tumor necrosis factor superfamily 14) is a protein expressed in activated T cells and NK cells, and may also be an interesting protein to study further, given its effects on CD8 cells’ entry into tumors [36]. Using a murine mammary tumor model, Dai et al. injected a human adenoviral vector expressing TNSFN14, and demonstrated a strong antitumor response [49]. We observed decreased levels of circulating TNFSF14 3 months after the termination of treatment, which might indicate a downregulation after the resolved disease/activation of the adaptive immune response.


*CD244*


CD244 is a transmembrane immunomodulatory receptor, and has been detected in NK cells and subsets of CD8+ T cells [50]. CD244 can be classified as an immunosuppressive receptor—as shown in a recent study on a murine model of colon cancer—and, as such, can contribute to the immunosuppressive TME [51]. Based on the findings of tumor growth studies in a murine model of HPV-driven HNSCC, CD244 has been suggested as a target for immunotherapy [37]. The finding of lower expression of this protein in the CRT group might indicate that cisplatin, in cooperation with RT, is involved in breaking down the immunosuppressive environment around the tumor.

The present study highlights a number of interesting inflammatory proteins with diverse biological activities associated with treatment response. Considering the myriad of roles that these proteins play in the immune response, a valuable critical insight of this study is that a high number of observed changes may reflect mechanisms that are not directly associated with cancer. Instead, it can be assumed that some of these proteins are also linked to inflammatory processes in host cells outside the tumor bed. Three confounders were identified in the present study, as our data demonstrated that mucositis, weight loss, and smoking pack-years impacted levels of IL-6 and IL-10, whereas smoking pack-years had a strong influence on IL-18 serum levels, corroborating some earlier findings [52,53].

Cytokine levels in HNC have been studied in several previous works using different analytical techniques [16,54]. Dickinson et al. used mass spectrometry to differentiate proteins in sera from HPV-positive and HPV-negative patients, and found that three proteins were expressed differently [55]. An extended version of the OLINK immuno-oncology panel has been used previously for the analysis of protein expression in fresh-frozen tissues from oropharyngeal cancer [56]. The authors identified several proteins that were related to disease recurrence, the most notable being DLL1, ESM-1, and EGF. An important finding in their study was the diversity of protein expression between tumor and normal tissues, which was also demonstrated in another study where tumors and stroma at different sub-locations of oral cancer were analyzed [57]. Recently, Mytilneous et al. described the evolution of a cytokine panel during and after treatment for HNC [54]. A different panel was used, but the authors found that patients with HPV-positive tumors had higher levels of pro-inflammatory cytokines in their sera, likely reflecting the immunogenic nature of HPV-positive disease. Apart from our group’s previously published study, this is, to the best of our knowledge, the only published paper to describe such a longitudinal analysis [16].

Several studies report the effects of treatment on serum biomarkers in cancer patients [58,59]. The results obtained in the present study may provide information on the therapeutic effects of concomitant cisplatin treatment in patients with HNC, and highlight opportunities for further research on cisplatin treatment in patients with HNC. The mechanisms that underlie altered protein expression at the end of treatment need to be explored further, and may also open up new perspectives in the study of resistance to cisplatin, which is a major clinical problem.

The strengths of this study include the large size of the cohort and the number of proteins analyzed. The main weakness of the study is the heterogeneous nature of the cohort, which is common in HNC research; however, at the same time, this can also be viewed as a strength, as it provides an opportunity to compare different treatment modalities within the study cohort. In the context of HNC immunology, the association between serum levels and the underlying molecular events in the tumor compartments is important to identify in future studies.

## 4. Materials and Methods

### 4.1. Study Population

The patients were prospectively recruited at three tertiary head and neck centers in Sweden. Inclusion criteria were newly diagnosed, curable, untreated HNC with a performance status of 0 to 2 according to the ECOG/World Health Organization (WHO) classification [60]. Exclusion criteria included previous treatment for malignant neoplasms within the last 5 years (except for skin cancer), excessive alcohol use, cognitive impairment, and inability to understand Swedish. The Uppsala Regional Ethics Review Board reviewed and approved the study (No. 2014/447). Blood samples were coded and stored in the Uppsala Biobank (approved RCC 2015-0025). This study was carried out as part of a larger prospective study registered on ClinicalTrials.gov, NCT03343236.

All patients were under nutritional surveillance according to local protocols, and supplementary nutritional therapy was offered when indicated. All patients were classified according to the Union for International Cancer Control (UICC) 8 staging system. A study representative met with the included patients and collected blood samples before treatment, 7 weeks after the start of treatment, and 3 and 12 months after the end of treatment. Body weight was monitored at each visit, and the grade of mucositis was evaluated according to the WHO mucositis scale [61]. Height was measured and used for the calculation of body mass index (BMI) (kg/m^2^). Any history of smoking was recorded and documented as pack-years.

The study cohort consisted of 180 patients with HNC. The most common sites were the oropharynx (*n* = 81), the oral cavity (*n* = 53), and the larynx (*n* = 22). Patient characteristics are shown in Table 3. Data from the first 30 patients in this study have previously been reported [16].

In terms of treatment, the patients were divided into 4 groups:(a)Surgery only (Surg group), *n* = 24.(b)RT +/− surgery (RT group), *n* = 94.(c)RT and chemotherapy (cisplatin) +/− surgery (CRT group), *n* = 47.(d)RT and targeted therapy (EGFR monoclonal antibody) +/− surgery (RT Cetux group), *n* = 15.

Radiation therapy was administered with conventional fractionation (2 Gy/fraction), up to 68–70 Gy for primary therapy and 60–70 Gy for adjuvant therapy.

Cisplatin was administered concomitantly with radiation therapy to a total of 47 patients, in weekly doses of 40 mg/m^2^: 45 patients with oropharyngeal cancer, and 2 patients with laryngeal cancer. Most patients received 5 to 8 courses of cisplatin (*n* = 34), and the remainder received 1 to 4 courses (*n* = 13). In addition, 15 patients received weekly cetuximab (Cetux) and concomitant RT. Cetux was administered at a loading dose of 400 mg/m^2^ and then a weekly dose of 250/m^2^ in most cases. Two patients received 9 doses, five patients received 8 doses, four patients received 7 doses, and the remainder received 4–6 doses.

A total of 6 patients received brachytherapy in addition to conventional RT.

### 4.2. Immune Marker Measurement

Blood was drawn from all patients at regular intervals (before treatment, 7 weeks after the start of treatment, and 3 and 12 months after the end of treatment) and stored as serum at −70 °C at the Uppsala University Hospital Biobank, Uppsala. Sera were thawed and clarified via centrifugation (2000× *g* 10 min). The clarified samples were then transferred to 96-well plates, which were analyzed using an immuno-oncology biomarker panel (Olink, Uppsala, Sweden) [62]. This panel offers a simultaneous multiplex immunoassay analysis of 92 protein biomarkers that are involved in key biological processes such as adaptive immune response, lymphocyte activation, inflammatory response, and cytokine-mediated signaling pathways. This panel has previously been used in numerous studies on biomarkers and cancer-related inflammation [63,64,65]. The panel has undergone validation both for single antibody pairs and as a full panel [62]. For the specific validation of this panel, please see, (https://www.olink.com/content/uploads/2021/09/olink-immuno-oncology-validation-data-v2.1.pdf, accessed on 1 September 2021) [66]. The results were delivered as normalized protein expression (NPX)—an arbitrary unit at Log2 scale. Membrane-bound proteins were measured in their soluble forms. From the immuno-oncology panel, 9 proteins were excluded from further analysis (HGF, VEGFA, VEGFC, VEGFR-2, PGF, TGF-β, PDGF, FGF-2, and EGF), leaving 83 proteins for analysis. For a complete list of the proteins analyzed, please see Appendix A.

### 4.3. T-Distributed Stochastic Neighbor Embedding (tSNE)

As shown in Figure 3, tSNE was used to verify that samples originating from the same patient clustered together. Samples from the same patients were expected to cluster together, as it can be hypothesized that these would be more similar for each individual patient. The tSNE indicated that the samples were suitably clustered.

### 4.4. Statistical Analysis

The raw data were first processed via Olink statistical analysis support, and 2 samples were identified as outliers and excluded from further analysis. T-distributed stochastic neighbor embedding (tSNE) analysis was then performed to verify that samples from the same patient were clustered together; the perplexity was set to 10 and 5000 iterations.

Figure data are presented as means ± standard deviation (SD). Significant differences (*p*-value < 0.05) are denoted with *, and the number of * signifies the level of significance (<0.05 = *, 0.01 = **, <0.001 = ***). The statistics in the text are reported as *p*-values, mean differences between tested groups, and 95% confidence intervals (CIs) (where possible), in the format of *p* = 0.05, z% (x–y), where z is the mean difference between the groups, and x and y are the limits of the 95% CI. Changes over time in circulating protein expression levels were calculated using two-way ANOVA, comparing time point 1 with all other time points. The analysis was adjusted for multiple comparisons using Dunnett’s multiple comparisons test.

For treatment effects, a three-stage approach was used. First, two-way ANOVA was performed for the time point after the completion of treatment (time point 2), comparing all proteins between the four treatment modalities. The analysis was adjusted for multiple comparisons using Dunnett’s multiple comparisons test. Then, the differences in protein expression levels between the groups (*n* = 17) were further analyzed using a linear regression model including the variables smoking, weight change, staging, treatment, and mucositis. When treatment predicted the change in protein expression levels (*n* = 14), the result was considered a discovery, and was compared between the RT group and the other treatment groups using one-way ANOVA, with Dunnett’s multiple comparisons test being used to calculate the differences between the groups. When another variable (i.e., pack-years, weight change, stage, or mucositis) predicted the change in protein expression levels, the result was considered a non-discovery, and was not analyzed further (*n* = 3). For the complete results of the linear regression, please see the Appendix A.

## 5. Conclusions

In this longitudinal real-world study of patients undergoing treatment for HNC, there was a significant association between serum inflammatory proteins and treatment modality. Compared to RT with or without surgical invention, RT with concomitant cisplatin induced a significant decrease in 13 of the 83 serum proteins analyzed by the end of treatment. To date, few studies have analyzed the effects of chemoradiation on circulating inflammatory biomarkers. This longitudinal analysis draws attention to several inflammatory proteins that are of interest for further exploration.

## Figures and Tables

**Figure 1 ijms-23-06304-f001:**
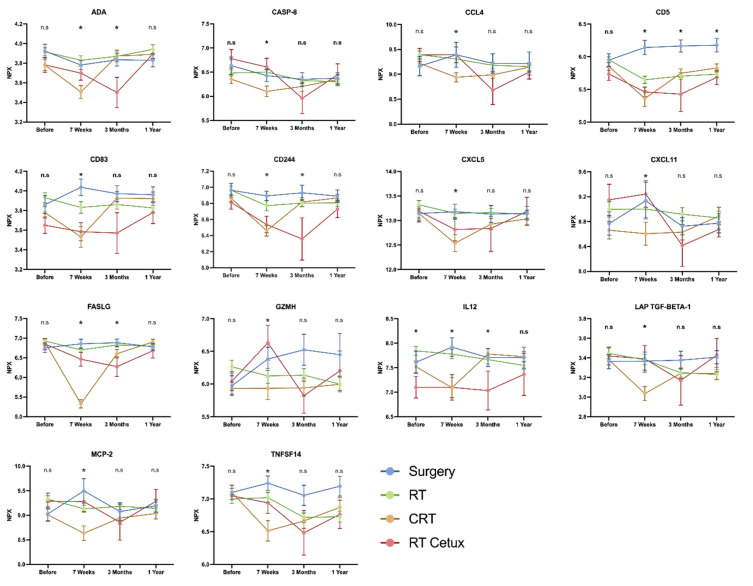
The longitudinal changes of 14 inflammatory proteins presented for each treatment group. Proteins exhibiting significant differences between treatment groups at the three follow-up time points are indicated, * denotes that there were statistically significant differences between the treatment groups.

**Figure 2 ijms-23-06304-f002:**
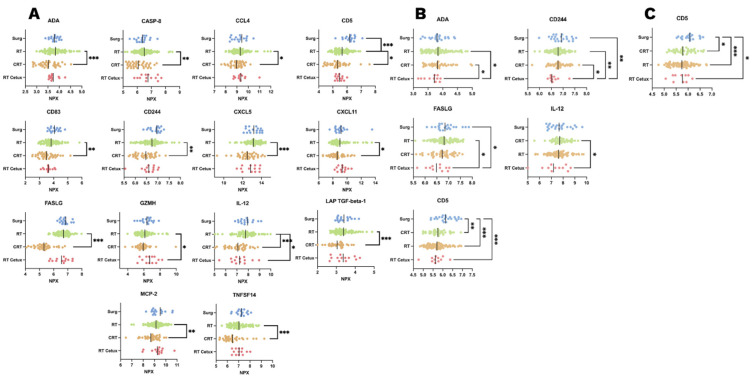
Proteins exhibiting significant differences between treatment groups at 7 weeks after the start of treatment for head and neck cancer (**A**), 3 months after the termination of treatment (**B**), and 12 months after the termination of treatment (**C**). (* *p* ≤ 0.05; ** *p* ≤ 0.01; *** *p* ≤ 0.001).

**Figure 3 ijms-23-06304-f003:**
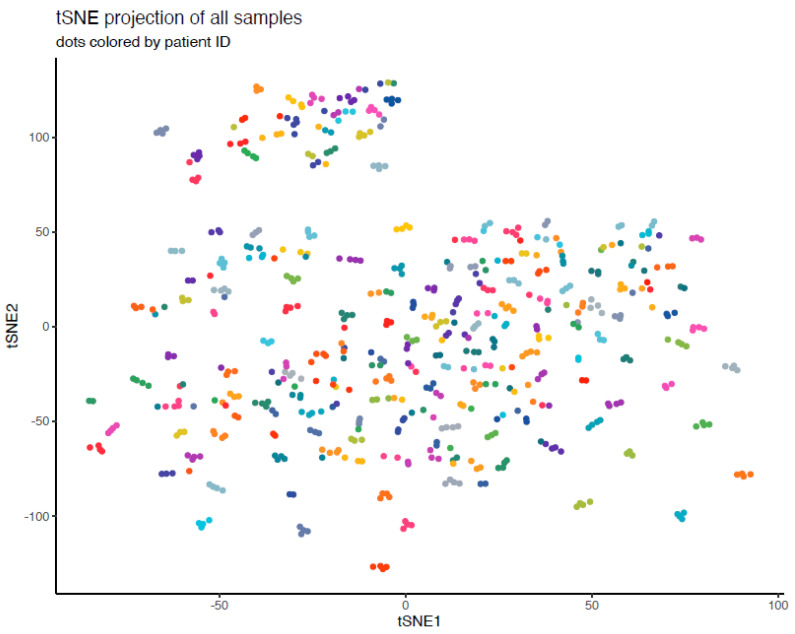
tSNE projection of the dataset after omission of outliers and samples failing quality control.

**Table 1 ijms-23-06304-t001:** Proteins that exhibited significantly increased or decreased expression at 7 weeks after the start of treatment, and at 3 months and 12 months after the termination of treatment, compared with baseline.

7 Weeks		3 Months		12 Months	
Decreased	Increased	Decreased	Increased	Decreased	Increased
ARG1 *	CCL17 ***	ARG1 ****	CCL17 **	ARG1 **	CCL17 *
CCL20 ****	CCL23 *	CASP-8 *		CCL20 ****	
CD5 ***	IL-6 ****	CCL4 *		CXCL13 *	
CD244 **	IL-7 *	CCL19 *		TNFSF14 *	
CXCL5 ***	IL-10 ****	CCL20 ****			
FASL ****		CXCL13 **			
IL-13 **		MMP12 *			
LAMP3 *		TNF *			
MCP-4 *		TNFSF14 ***			
MMP12 ****					

* *p* ≤ 0.05; ** *p* ≤ 0.01; *** *p* ≤ 0.001; **** *p* ≤ 0.0001.

**Table 2 ijms-23-06304-t002:** Correlation between differences in protein expression and mucositis, tumor stage, smoking pack-years, and weight change, according to multivariate analysis. Proteins in bold indicate stronger correlation with the variable than the treatment.

*Mucositis*	*Tumor Stage*	*Pack-Years*	*Weight Change*
**IL-6 (*p* = 0.0004)**	GZMH (*p* = 0.0086)	CD5 (*p* = 0.0145)	IL-6 (*p* = 0.0038)
IL-10 (*p* = 0.0479)		CD83 (*p* = 0.0134)	IL-10 (*p* = 0.0381)
		IL-6 (*p* = 0.0016)	
		**IL-10 (*p* = ** **0.038)**	
		**IL-18 (*p* = ** **0.0014)**	

**Table 3 ijms-23-06304-t003:** Patient characteristics.

**Characteristics**	**n (%)**
**Number of patients**	180
Female	48 (26.7)
Male	132 (73.3)
**Age (in years)**	
Mean	62.8
Range	34–85
**Smoking habits**	
Never	57 (31.7)
Previous	107 (59.4)
Active	16 (8.9)
**Pack-years**	
1–20	60
21–50	27
>50	15
Missing	21
**Tumor site**	
Oropharynx	81 (45)
p16-positive	77
p16-negative	4
Oral cavity	53 (29.4)
Larynx	22 (12.2)
Sinonasal	6 (3.3)
Hypopharynx	5 (2.7)
Salivary gland	4 (2.2)
Nasopharynx	3 (1.7)
CUP ^a^	3 (1.7)
Other ^b^	3 (1.7)
**Stage ^c^**	
I	77 (42.8)
II	35 (19.4)
III	29 (16.1)
IVa	32 (17.8)
IVb	6 (3.3)
IVc	1 (0.6)
**Treatment**	
RT +/− surgery	94 (52.2)
CRT ^d^	47 (26.1)
RT + targeted therapy ^e^	15 (8.3)
Surgery only	24 (13,4)
**Mucositis grade ^f^**	
0	34
1	22
2	35
3	55
4	8

^a^ Cancer of unknown primary; ^b^ external auditory canal cancer; ^c^ according to UICC 8; ^d^ radiotherapy + cisplatin; ^e^ radiotherapy + cetuximab; ^f^ at 7 weeks after the start of treatment.

## Data Availability

Data are available upon request.

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
