# Peer review of "Serum Proteomics in Patients with Head and Neck Cancer: Peripheral Blood Immune Response to Treatment"

_ijms, 2022, doi:10.3390/ijms23116304_

Round 1
Reviewer 1 Report
Astradsson and colleagues present a well-written manuscript evaluating the expression of inflammatory proteins in serum of head and neck cancer patients before and after treatment. Although no mechanistic insights that underlie the altered protein expression are investigated, it does give novel insight into the longitudinal expression of proteins that play a major role in the (anti-) tumor immune response. It thereby highlights specific proteins that could be of interest for further (more mechanistic) studies.
I do have some remarks that need to be addressed before I would recommend acceptance in IJMS:
1) I miss an overview of all the proteins analyzed + a rationale for the choice of these proteins. The analyzed proteins should be provided in a table.
In the M&M section 4.2 it is mentioned that 9 proteins were excluded from analysis. The authors should provide a reason for that.
2) Table 1: the legend could be a bit more clear stating that all timepoints are compared to baseline
3) Figure 2: why are these particular proteins shown? In the legend it states that “proteins exhibiting significant differences between treatment groups” are indicated. But I also see protein graphs without any significance. And the text states that there are 17 proteins that exhibit significant changes, while 14 proteins are shown. A better rationale should be given in the text why some proteins are and some proteins are not shown.
4) Figure 2: From this figure it is now clear that there is “a” significant difference between treatments, but not between which treatments there is a significant difference. A different representation of the data would give more useful insight into the significant differences of protein expression between specific treatment groups.
5) M&M 4.2: More details should be provided regarding the processing of the serum and the Olink procedure.
Reviewer 2 Report
Comments to the Authors
The authors utilize an immuno-oncology biomarker panel consisting of 83 antibodies to profile serum samples from tumor samples from 180 patients with head and neck cancer. Pretreatment, 7-we following treatment initiation, 3 month FU and 12 month FU samples were investigated. Interestingly, cisplatin-based chemoradiation has immunological effects as well as the majority of the inflammatory proteins had returned to the pretreatment level at 12 months FU. Generally, the background information is adequate and informative as well as well-written, describing the current knowledge of protein profiling in head and neck cancer. The methodology and results are generally well described.
The following requires attention:
1) How is the immuno-oncology biomarker panel validated. Please specify in the text how the antibodiesare validated – specificity? Are the antibodies validated in single assays – or in combination (all 83) as used in the panel in this manuscript? Please provide more information on this pivotal basis for the methodology used or include reference if such data is already published elsewhere.
2) In the result section, please rethink if ‘2.1. tSNE’ should be in the M&M section. Is this a result or just your quality control?
How did you determine that the samples from each patient clustered together? You describe that these is optimally 4 samples from each patient, but when just by eye looking af the tSNE, many samples (same colors) do not cluster. Please describe this decision of the samples from each patient cluster and/or show this in a Figure that illustrates which patient samples that cluster and which does not.
3) This this is a probably a typo: In section 2.2., Table 2 is mentioned prior to Table 1 in the text. However, ‘as shown in Table 2, for the whole cohort…” does not match with Table 2. Please correct or explain.
4) In Table, 1 please write the proteins in alphabetic order, or any order you decide to give the reader a more structured presentation.
5) Consistently, through the text you often use “protein changed...”. It is not the proteins that change but it is the expression levels of the proteins that are different/change. Please go through the text with this in mind.
6) Please bring your attention to provide reference for all your statements of ‘previous knowledge’, e.g. in the discussion section “Preclinical research, mainly in mouse models, has also been used to explore the immunogenic properties of cisplatin in combination with radiation therapy.” or “Other animal models have also demonstrated a potent antitumor effect of IL-12, but this has not been observed in humans so far”.Here, references should be provided.
Taken together the manuscript shows novelty in the field of immune response to treatment in Head and Neck cancer. It is generally well described and present findings that both extend and fit the contemporary knowledge. In conclusion, I find this study relevant in regard to the research field of pre-clinicallycharacterization. Validation of the results in independent cohorts as well as functional studies are awaited with great anticipation.
Round 2
Reviewer 1 Report
Although the resolution of the figures in the revised manuscript was very low, making the figures hard to read, I believe the authors have adequately addressed my remarks to support publication of this manuscript.